# GWAS of mosaic loss of chromosome Y highlights genetic effects on blood cell differentiation

Chikashi Terao [1,2,3]*, Yukihide Momozawa[4], Kazuyoshi Ishigaki[1], Eiryo Kawakami[5,6,7], Masato Akiyama[1,8], Po-Ru Loh [9,10], Giulio Genovese[10,11,12], Hiroki Sugishita[13], Tazro Ohta [14], Makoto Hirata [15], John R.B. Perry[16], Koichi Matsuda [15,17], Yoshinori Murakami [18], Michiaki Kubo[4] & Yoichiro Kamatani [1,19]*

Mosaic loss of chromosome Y (mLOY) is frequently observed in the leukocytes of ageing men. However, the genetic architecture and biological mechanisms underlying mLOY are not fully understood. In a cohort of 95,380 Japanese men, we identify 50 independent genetic markers in 46 loci associated with mLOY at a genome-wide significant level, 35 of which are unreported. Lead markers overlap enhancer marks in hematopoietic stem cells (HSCs, $P \leq 1.0 \times 10^{-6}$). mLOY genome-wide association study signals exhibit polygenic architecture and demonstrate strong heritability enrichment in regions surrounding genes specifically expressed in multipotent progenitor (MPP) cells and HSCs ($P \leq 3.5 \times 10^{-6}$). ChIP-seq data demonstrate that binding sites of FLI1, a fate-determining factor promoting HSC differentiation into platelets rather than red blood cells (RBCs), show a strong heritability enrichment ($P = 1.5 \times 10^{-6}$). Consistent with these findings, platelet and RBC counts are positively and negatively associated with mLOY, respectively. Collectively, our observations improve our understanding of the mechanisms underlying mLOY.

[1] Laboratory for Statistical and Translational Genetics, RIKEN Center for Integrative Medical Sciences, Kanagawa 230-0045, Japan. [2] Clinical Research Center, Shizuoka General Hospital, Shizuoka 420-8527, Japan. [3] The Department of Applied Genetics, The School of Pharmaceutical Sciences, University of Shizuoka, Shizuoka 422-8526, Japan. [4] Laboratory for Genotyping Development, RIKEN Center for Integrative Medical Sciences, Yokohama, Kanagawa 230-0045, Japan. [5] Healthcare and Medical Data Driven AI based Predictive Reasoning Development Unit, Medical Sciences Innovation Hub Program (MIH), RIKEN, Kanagawa 230-0045, Japan. [6] Laboratory for Developmental Genetics, Center for Integrative Medical Sciences (IMS), RIKEN, Yokohama, Kanagawa 230-0045, Japan. [7] Artificial Intelligence Medicine, Graduate School of Medicine, Chiba University, Chiba 260-8670, Japan. [8] Department of Ophthalmology, Graduate School of Medical Sciences, Kyushu University, Fukuoka 812-8582, Japan. [9] Division of Genetics, Department of Medicine, Brigham and Women's Hospital and Harvard Medical School, Boston, MA 02115, USA. [10] Program in Medical and Population Genetics, Broad Institute of MIT and Harvard, Cambridge, MA 02142, USA. [11] Department of Genetics, Harvard Medical School, Boston, MA 02115, USA. [12] Stanley Center for Psychiatric Research, Broad Institute of MIT and Harvard, Cambridge, MA 02142, USA. [13] Laboratory for Developmental Genetics, RIKEN Center for Integrative Medical Science (IMS), Yokohama, Kanagawa 230-0045, Japan. [14] Database Center for Life Science, Joint Support-Center for Data Science Research, Research Organization of Information and Systems, Mishima, Shizuoka 411-8540, Japan. [15] Laboratory of Genome Technology, Human Genome Center, Institute of Medical Science, The University of Tokyo, Tokyo 108-8639, Japan. [16] MRC Epidemiology Unit, School of Clinical Medicine, University of Cambridge, Cambridge CB2 0SP, UK. [17] Laboratory of Clinical Genome Sequencing, Department of Computational Biology and Medical Sciences, Graduate School of Frontier Sciences, The University of Tokyo, Tokyo 108-8639, Japan. [18] Division of Molecular Pathology, Institute of Medical Science, The University of Tokyo, Tokyo 108-8639, Japan. [19] Laboratory of Complex Trait Genomics, Department of Computational Biology and Medical Sciences, Graduate School of Frontier Sciences, The University of Tokyo, Tokyo 108-8639, Japan. *email: chikashi.terao@riken.jp; yoichiro.kamatani@riken.jp

Mosaic loss of chromosome Y (mLOY)[1] is characterized by the presence of cells that have lost chromosome Y. mLOY is well described in samples from the bone marrow[2] and can also be detected in peripheral blood cells[3]. Age[4] and smoking[5] are well established risk factors for mLOY, however, the broader mechanisms that influence mLOY, including the stage(s) of hematopoietic differentiation during which mLOY arises, are not fully known. Genetic analysis can provide biological insight into the basis of this phenomenon. The same array-based assays can provide information about single-nucleotide polymorphism (SNP) genotype and, from intensity data of genome-wide probes, allow inference of mosaic events[6,7]. This has allowed previous studies to identify some genetic determinants underlying mLOY. Zhou et al. reported that mLOY is associated with a variant in the *TCL1A* gene locus[8]. Wright et al. identified a total of 19 loci associated with mLOY[9], including *TCL1A* and other genes involved in cell cycle regulation and DNA damage response. Previous studies examined European populations, so genetic studies from an East Asian population would expand our knowledge of the genetic architecture underlying mLOY, in turn allowing us to make deeper biological inferences about mLOY based on expanded genetic findings.

The clinical significance of mLOY is still unclear[10]. Previous studies reported that mLOY is associated with shorter life span and increased risk for cancer onset[3,11,12]. Other studies suggested that mLOY is associated with onset of acute myeloid leukemia (AML) or other hematological disorders[13,14]. However, Zhou et al. did not find significant differences in overall survival or cancer-specific survival in data from 5340 cancer cases[8]. No studies have analyzed the clinical significance of mLOY in Asian populations. Although the clinical significance of mLOY is not well established, clarifying the mechanisms of chromosome loss may provide a deeper understanding of clonal expansion during hematopoiesis, cellular ageing, and cancer development. Furthermore, there is a possibility that mLOY has a distinct clinical impact in Asian populations.

Here, we use genotype array data from 95,380 Japanese males enrolled in the Biobank Japan Project (BBJ) to study mLOY[15]. We examine the genetic architecture of mLOY, performing the genome-wide association study (GWAS) of this condition in an East Asian population. To infer biological mechanisms driving mLOY, we conduct downstream analyses using our GWAS results, making inferences based on overlap of associated variants and polygenic mLOY heritability with functional elements to pinpoint the cell types and gene pathways involved in mLOY. Furthermore, we conduct association studies between mLOY and survival data of the participants to evaluate the clinical significance of mLOY in an Asian population.

## Results

**Estimation of mLOY from probe intensity data of DNA microarray.** The data were generated in three separate batches due to advances in genotyping platforms during data collection. Detailed description of the samples is shown in Supplementary Table 1.

We applied a method similar to that used in Wright et al.[9] to estimate mLOY in our data. We obtained logarithm of R ratio (LRR) probe intensity data across more than 1100 variants (depending on batch) in chromosome Y for each male subject. We used mean LRR (mLRR-Y) as a proxy for mean Y chromosome dosage in circulating blood cells of subjects (for details, see the Methods section). Hereafter, we use the term 'mLOY GWAS' to denote GWAS on mLRR-Y, as in Wright et al.[9].

**Associations between mLOY and age or smoking.** We observed a strong association between age at DNA collection and mLOY (1 year increase in age associated with 2.2% standard deviation (SD) decrease in mLRR-Y signal, $P < 1.0 \times 10^{-100}$, Wald test in linear regression analysis, Fig. 1a; Supplementary Fig. 1) explaining 9.6% of the variance in mLOY. We also observed a significant association between smoking and mLOY (Supplementary Fig. 2); smokers often have lower mean intensity of chromosome Y probes, indicating a higher fraction of cells with loss of chromosome Y (smokers associated with 4.6% SD decrease in mLRR-Y signal, $P = 7.5 \times 10^{-10}$, Wald test in linear regression analysis). These associations are in agreement with previous studies[5].

**mLOY as a highly polygenic trait.** Next, we imputed genotypes using genotyping array data after standard quality control (Methods) and a reference panel of whole-genome sequenced individuals, including 2504 individuals from the 1000 Genomes project phase 3 and 1,037 Japanese[16,17] (Methods

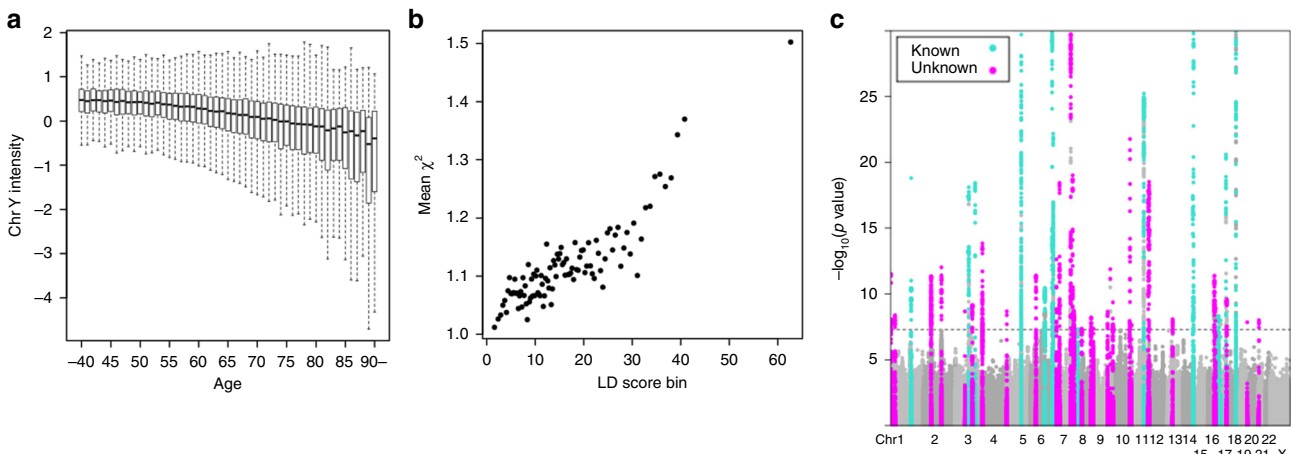

**Fig. 1** Associations of Chromosome Y signals with old age and a total of 50 genetic determinants in mLOY. **a** An association between mLOY and age at blood collection is indicated. We do not show outliers in the figure. Bars indicate the most extreme data points which are no more than 1.5 times interquartile ranges from the boxes. **b** Polygenetic architecture in mLOY. Mean chi-square statistics and LD scores in 100 bins of Hapmap project variants in mLOY are indicated. Bins are made according to LD scores to contain almost equal number of variants in each bin. **c** Manhattan plot of mLOY in this study. Green and red colors indicate previously reported and unreported regions, respectively. Only $P$-values $>1.0 \times 10^{-30}$ are shown

and Supplementary Note 1). We conducted a Bayesian mixed-model association study to identify mLOY susceptibility loci, controlling for age, smoking, genotyping batch, and disease status (for details, see the Methods section).

We found strong evidence for polygenicity of mLOY GWAS signals. Chi-squared statistics for association with mLOY deviated slightly from expected statistics in a quantile–quantile plot (lambda = 1.066, Supplementary Fig. 3). We evaluated polygenic effects on mLOY using linkage disequilibrium (LD) score regression[18], with the use of ldsc software. This regression analysis suggested that mLOY is a highly polygenic trait and that departure of mean chi-squared statistics could be largely explained by polygenic effects (lambda genomic control 1.086 > intercept 1.019 in ldsc, Fig. 1b and Supplementary Note 2). Since LD score regression revealed minimal bias, we did not correct study statistics by genomic control[19].

**Common genetic architecture of mLOY with Europeans.** To understand our results in the context of what is known about the genetics of mLOY in Europeans, we compared our results to previous analyses. We found a total of 46 loci significantly associated with mLOY, 15 of which were reported by Wright et al.[9] from analysis of data from the UK BioBank (Fig. 1c, Tables 1, 2; Supplementary Fig. 4). We found a consistent association trend (direction of effect of minor alleles) across all three batches for each of the 46 associated variants (Supplementary Table 2). Conditional analyses revealed that four loci, all of which were among those found in Wright et al.[9], contained two previously undetected independent signals (Table 3; Supplementary Fig. 5). The 50 independent variants over 46 loci explained 3.2% of mLOY variance, 1.6% of which could be explained by the 35

new variants. We estimated SNP-heritability of mLOY to be 9.3% (see the Methods section).

We did not find significant associations in 4 of the 19 regions identified by Wright et al.[9] (Supplementary Fig. 6). The most strongly associated variant (rs17758695) in the *BCL2* gene locus, the top locus identified by Wright et al.[9], is not polymorphic in the Japanese population (Supplementary Table 3). The top variants in the other three regions identified by Wright et al.[9], but not here, were also very rare or not polymorphic in the Japanese population (Supplementary Table 3). Therefore relative invariance at these four loci in the Japanese population precludes identification of a statistically significant signal in this population.

To further assess similarities and differences in the genetic architecture of mLOY across populations, we examined associations of the 46 loci with mLOY in 205 thousand male participants in the full UK Biobank release (Supplementary Table 4). Thirty-nine of the 46 top variants were polymorphic, and had minor allele frequencies (MAF) >0.5% in the UK population. Thirty-seven of these 39 variants had the same effect direction in UK Biobank and BBJ ($P = 2.8 \times 10^{-9}$, binomial test). When considering only the 31 novel loci, this trend remained consistent; twenty-five variants were polymorphic and had MAF >0.5% in the UK population, and 23 out of the 25 variants had the same effect direction in the two studies ($P = 1.9 \times 10^{-5}$, binomial test), indicating strong genetic overlap in association with mLOY between the two populations.

**Involvement of hematopoietic stem cells with mLOY.** In order to assess the cell types important for mLOY based on significant GWAS signals, we next analyzed the 46 lead variants in BBJ for overlap with cell-type-specific enhancer marks using Haploreg[20].

**Table 1 A total of 31 previously unreported significant loci in mLOY**

| SNP | Chr | BP | Gene | A0 | A1 | A1frq | Beta | P |
|---|---|---|---|---|---|---|---|---|
| rs34468831 | 1 | 3097312 | PRDM16 | GA | G | 0.37 | −0.032 | $3 \times 10^{-12}$ |
| rs527504 | 1 | 33392427 | TMEM54;RNF19B | G | A | 0.19 | −0.032 | $4.1 \times 10^{-9}$ |
| rs17049722 | 2 | 58976863 | LINC01122 | C | T | 0.17 | 0.038 | $4.4 \times 10^{-12}$ |
| 2: 136879065 | 2 | 136879065 | CXCR4 | G | ALU | 0.32 | −0.035 | $9.3 \times 10^{-13}$ |
| rs34778241 | 3 | 71771215 | EIF4E3 | T | TG | 0.71 | −0.030 | $2.1 \times 10^{-9}$ |
| rs2811487 | 3 | 128331879 | LINC01565;RPN1 | G | A | 0.36 | 0.027 | $6.8 \times 10^{-10}$ |
| rs871134 | 4 | 7044380 | CCDC96 | C | T | 0.49 | −0.032 | $1.4 \times 10^{-14}$ |
| rs2853677 | 5 | 1287194 | TERT | G | A | 0.70 | 0.027 | $2.1 \times 10^{-9}$ |
| rs10948011 | 6 | 42024285 | TAF8 | G | A | 0.23 | 0.035 | $4 \times 10^{-12}$ |
| rs35355140 | 7 | 27204732 | HOXA9 | C | A | 0.14 | 0.045 | $2.3 \times 10^{-13}$ |
| rs11769630 | 7 | 50257703 | C7orf72;IKZF1 | T | A | 0.13 | −0.057 | $3.7 \times 10^{-19}$ |
| rs59543286 | 7 | 135351310 | C7orf73 | A | C | 0.23 | 0.059 | $8 \times 10^{-32}$ |
| rs55727837 | 7 | 149428602 | KRBA1 | G | T | 0.11 | 0.061 | $9.7 \times 10^{-20}$ |
| rs12668837 | 7 | 158500805 | NCAPG2;ESYT2 | C | T | 0.46 | −0.026 | $1.8 \times 10^{-9}$ |
| rs189309686 | 8 | 59509355 | NSMAF | C | T | 0.25 | 0.027 | $4.4 \times 10^{-8}$ |
| rs10692222 | 8 | 130597362 | CCDC26 | C | CATT | 0.43 | 0.025 | $6 \times 10^{-9}$ |
| rs2804301 | 9 | 603916 | KANK1 | G | A | 0.24 | −0.027 | $4.4 \times 10^{-8}$ |
| rs9299129 | 9 | 109638167 | ZNF462 | A | G | 0.23 | 0.030 | $2.2 \times 10^{-9}$ |
| rs138423884 | 9 | 129855937 | ANGPTL2;RALGPS1 | A | G | 0.038 | 0.079 | $1.3 \times 10^{-12}$ |
| rs2646425 | 10 | 8470387 | LINC00708;LOC105376398 | C | T | 0.31 | 0.028 | $4.3 \times 10^{-10}$ |
| rs12225799 | 11 | 241124 | PSMD13 | C | G | 0.13 | −0.036 | $1.1 \times 10^{-8}$ |
| rs2237896 | 11 | 2858440 | KCNQ1 | G | A | 0.40 | 0.043 | $1.7 \times 10^{-22}$ |
| rs74843651 | 11 | 14292987 | SPON1;RRAS2 | G | T | 0.052 | 0.055 | $2.2 \times 10^{-8}$ |
| rs10849448 | 12 | 6493351 | LTBR | A | G | 0.81 | −0.050 | $1.5 \times 10^{-18}$ |
| rs34324 | 12 | 12877926 | APOLD1 | C | A | 0.43 | −0.039 | $3.1 \times 10^{-19}$ |
| rs548509555 | 13 | 57601900 | MIR5007;PRR20B | T | TA | 0.20 | 0.032 | $8.1 \times 10^{-9}$ |
| rs9921295 | 16 | 50027130 | ZNF423;CNEP1R1 | T | G | 0.33 | 0.031 | $4 \times 10^{-12}$ |
| rs1859259 | 16 | 57596552 | GPR114 | C | T | 0.58 | −0.025 | $1.1 \times 10^{-9}$ |
| rs77406149 | 17 | 53076247 | STXBP4 | A | G | 0.20 | 0.035 | $2.5 \times 10^{-10}$ |
| rs79058858 | 19 | 49979789 | FLT3LG | C | T | 0.0052 | 0.16 | $1.4 \times 10^{-8}$ |
| rs117587217 | 21 | 16631171 | NRIP1;USP25 | T | C | 0.016 | −0.10 | $9.7 \times 10^{-9}$ |

*Chr* chromosome, *BP* base pair position in NCBI build 37, *Gene* italicized gene name, *A1frq* A1 allele frequency, *Beta* A1 allele beta

**Table 2 Significant signals in the 15 known loci**

| SNP | Chr | BP | Gene | A0 | A1 | A1frq | Beta | P |
|---|---|---|---|---|---|---|---|---|
| rs2842873 | 1 | 156204653 | *PMF1;PMF1-BGLAP* | C | T | 0.37 | −0.056 | $1.5 \times 10^{-36}$ |
| rs4683900 | 3 | 101134696 | *SENP7* | C | T | 0.73 | 0.042 | $7.9 \times 10^{-19}$ |
| rs4681200 | 3 | 150002138 | *LINC01214* | C | G | 0.92 | 0.072 | $3.8 \times 10^{-19}$ |
| rs56116444 | 5 | 111061847 | *STARD4-AS1* | T | G | 0.31 | −0.086 | $9.2 \times 10^{-81}$ |
| rs11251 | 6 | 109689907 | *CD164* | G | T | 0.54 | 0.028 | $3.4 \times 10^{-11}$ |
| rs4709819 | 6 | 164463355 | *LOC102724152;MEAT6* | G | A | 0.41 | 0.049 | $1.3 \times 10^{-30}$ |
| rs4721217 | 7 | 1973579 | *MAD1L1* | C | T | 0.42 | −0.043 | $6.8 \times 10^{-23}$ |
| rs2979469 | 8 | 30285091 | *RBPMS* | G | C | 0.88 | −0.035 | $4.9 \times 10^{-8}$ |
| rs227079 | 11 | 108248686 | *C11orf65* | C | A | 0.48 | −0.044 | $5.9 \times 10^{-26}$ |
| rs728739 | 14 | 96182062 | *TCL1A;TUNAR* | A | G | 0.055 | 0.081 | $3.8 \times 10^{-18}$ |
| rs72698721 | 14 | 101181189 | *LINC00523;DLK1* | G | A | 0.32 | 0.055 | $4.2 \times 10^{-33}$ |
| rs77874075 | 16 | 81066339 | *CENPN* | T | G | 0.22 | −0.030 | $4.6 \times 10^{-9}$ |
| rs201753350 | 17 | 7579705 | *TP53* | C | T | 0.0062 | −0.15 | $1 \times 10^{-8}$ |
| rs78997619 | 17 | 47787161 | *FAM117A* | C | T | 0.060 | 0.083 | $2.5 \times 10^{-21}$ |
| rs80277818 | 18 | 42017901 | *LINC01478* | A | G | 0.26 | −0.075 | $7.8 \times 10^{-55}$ |

*Chr* chromosome, *BP* base pair position in NCBI build 37, *Gene* italicized gene name, *A1frq* A1 allele frequency, *Beta* A1 allele beta

**Table 3 Secondary signals in the four known loci**

| SNP | Chr | BP | Gene | A0 | A1 | A1frq | Beta | P |
|---|---|---|---|---|---|---|---|---|
| rs9487023 | 6 | 109590004 | *C6orf183* | G | A | 0.10 | 0.040 | $3.5 \times 10^{-8}$ |
| rs4251697 | 12 | 12874462 | *CDKN1B* | A | G | 0.081 | 0.059 | $1.5 \times 10^{-10}$ |
| rs139012944 | 14 | 96180705 | *TCL1A* | T | C | 0.071 | −0.051 | $3.8 \times 10^{-8}$ |
| rs8088824 | 18 | 42151261 | *LINC01601;SETBP1* | T | C | 0.82 | −0.056 | $3.4 \times 10^{-23}$ |

*Chr* chromosome, *BP* base pair position in NCBI build 37, *Gene* italicized gene name, *A1frq* A1 allele frequency, *Beta* A1 allele beta

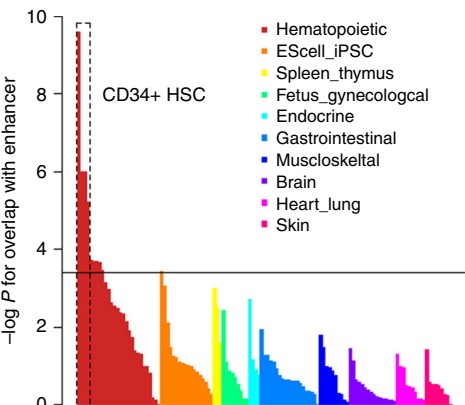

**Fig. 2** Significant variants in mLOY showing enrichment of enhancer marks in CD34+ HSCs. The results of enhancer enrichment of mLOY 46 top markers in 127 cell types calculated by Haploreg are indicated. Cell groups are shown in different colors. The results are sorted by cell groups and *P*-values. The horizontal solid line indicates significant level of *P*-value based on Bonferroni's correction (empirical *P* = 0.000394 (0.05/127))

We found a strong overlap between the variants and enhancer marks present in hematopoietic stem cells (HSCs) (Fig. 2; Supplementary Table 5). This finding is in line with the theory that clonal expansion of HSCs lacking chromosome Y underlies mLOY, as has been shown for clonal hematopoiesis of HSCs bearing a point mutation[21]. The 31 new loci found in the current study accounted for this overlap. Two completely independent sets of variants (31 and 19 loci in Japan and UK, respectively) showed overlap with enhancer marks in HSCs (Supplementary

Fig. 7). Enhancer marks in embryonic stem cells (ES cells) or induced pluripotent stem cells (iPS cells), both of which can differentiate into HSCs[22], did not strongly overlap with mLOY GWAS significant signals. Thus the variants associated with mLOY overlap with cell type-specific epigenetic features emerging after differentiation into hematopoietic cells, rather than epigenetic features characterizing pluripotency itself.

To further infer the functional implications of our genetic results regarding mLOY, we next analyzed the polygenic architecture of mLOY, not restricting to GWAS significant signals. We partitioned heritability of mLOY according to functional annotations[23] using ldsc (see the Methods section). Superenhancer, H3K27ac, and transcription start site (TSS) were the top three categories with positive and significant heritability enrichment (>2.5-fold enrichment, $P \leq 3.1 \times 10^{-6}$, Supplementary Table 6), suggesting that gene expression regulation, especially by superenhancers, is involved in the mechanisms underlying mLOY. Deep involvement of superenhancers with gene expression regulation may characterize mLOY genetic architecture, considering low enrichment of superenhancers in other traits' heritability[23]. We next conducted cell-group-specific heritability enrichment analysis using ldsc. This revealed that mLOY heritability was highly enriched in histone marks associated with the hematopoietic cell group (Fig. 3a). We next used ldsc to further analyze whether histone marks enriched in specific hematopoietic cell types, among a total of 220 cell types, could explain the heritability of mLOY. We found that histone marks specific for CD34-positive cells (HSCs and hematopoietic progenitor cells) showed the strongest heritability enrichment (Fig. 3b; Supplementary Table 7). The mLOY heritability enrichment in superenhancers, histone marks in the hematopoietic cell group, and CD34-positive cells were confirmed in the analyses with ldsc where we used different baseline annotations

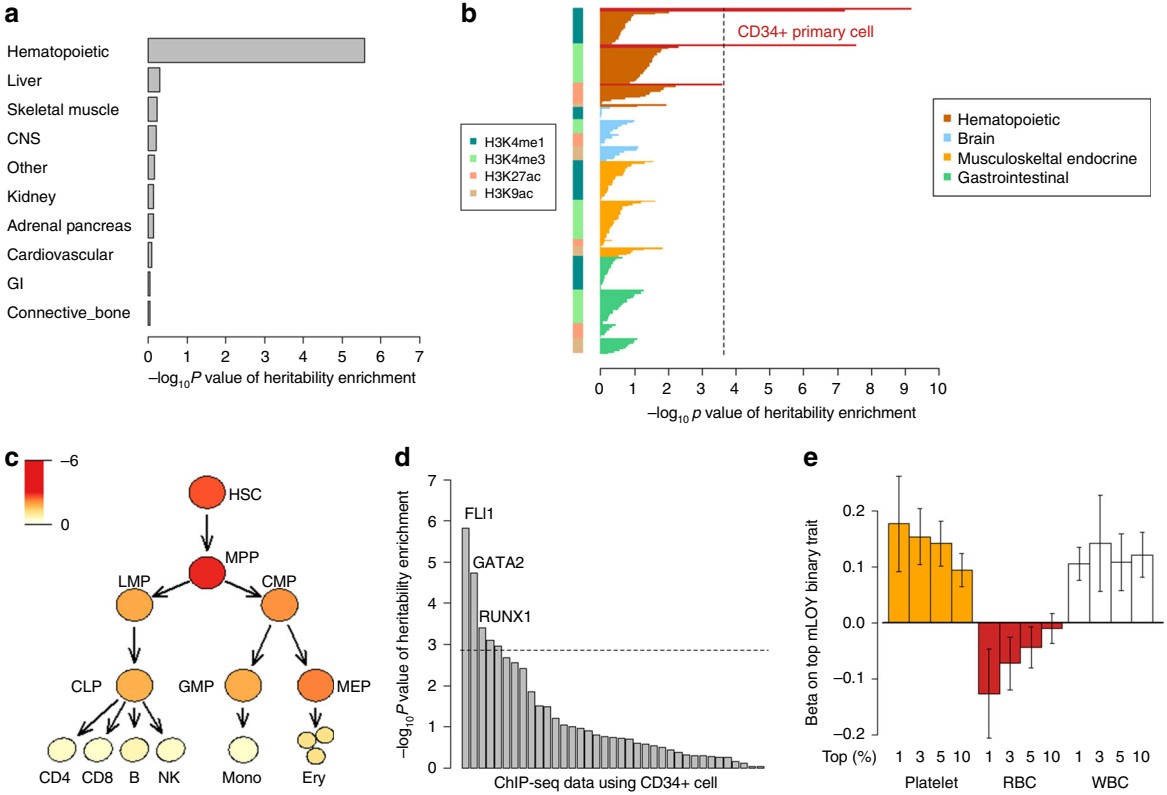

**Fig. 3** Involvement of CD34[+] HSCs and candidates of critical proliferation stage and transcription factor in mLOY. **a** Heritability enrichment of mLOY in ten cell groups evaluated by LDSC. The results of partitioning heritability for ten cell type groups with the use of ldsc are indicated. LD scores of the ten cell type groups are made based on LD scores of the fine cell types by the Finucane et al. The results are sorted by *P*-values. *P*-values of coefficient in ldsc are indicated for **a**–**d**. **b** Heritability enrichment of mLOY in 220 cell type-specific annotations evaluated by LDSC. The cell types are classified into cell groups accordingly. CD34[+] primary cells (HSCs) are given red color. A broken line indicates a significant threshold based on Bonferroni's correction (*P* < 0.05/ 220 cell type-specific annotations). The results are sorted by (1) cell groups, (2) annotations, and (3) *P*-values. **c** Heritability enrichment of mLOY in detailed hematopoietic cell lineages, including CD34[+] cells by LDSC-SEG. All of the cell types from HSC to CLP, GMP, MEP express CD34. The schema of differentiation from HSC is based on Corces et al. Brightness of the red color corresponds to strength of associations based on *P*-values (log10P). HSC hematopoietic stem cell, MPP multipotent progenitor, LMP lymphoid-primed multipotent progenitor, CMP common myeloid progenitor, CLP common lymphoid progenitor, GMP granulocyte/macrophage progenitor, MEP megakaryocyte/erythrocyte progenitor, CD4 CD4[+] T cell, CD8 CD8[+] T cell, B B cell, NK natural killer cell, Mono monocyte, Erythro erythrocyte. **d** Heritability enrichment of mLOY in transcription factor with the use of the ChIP-seq data. The results of partitioning heritability with the use of ldsc for ChIP-seq data where CD34[+] cells are used are indicated. The results are sorted by *P*-values. The broken horizontal line indicates a significant level based on Bonferroni's correction. The fourth and fifth significant data were again RUNX1 and GATA2 in different data sets, respectively. **e** Increased effect size of platelet count in association with mLOY according to mLOY fraction. Associations between individuals with top fractions of mLOY and counts of RBC, WBC, and platelets are analyzed in logistic regression analyses. Error bars indicate 95% confidence intervals

recently proposed to take into consideration difference in LD pattern of variants, information of synonymous/nonsynonymous coding variants, sequence age and conservation of variants across species[24] (see Methods and Supplementary Note 3). Taken together, both genome-wide significant loci and polygenic signals suggested involvement of CD34-positive cells with mLOY.

**Heritability enrichment in hematopoietic progenitor cells.** CD34 is expressed in hematopoietic progenitor cells, which include multipotent progenitor (MPP) and common myeloid progenitor (CMP), as well as HSC[25]. Therefore, to understand which differentiation stages are important for developing mLOY, we analyzed enrichment of mLOY heritability in chromatin regions that are open in various hematopoietic progenitor cells and lineages. We analyzed the GWAS results of mLOY with ldsc applied to specifically expressed genes (LDSC-SEG) using ATAC-seq data of hematopoietic progenitor cells and lineages at different differentiation stages[26]. Genes expressed in MPP and HSC, evidenced by

open-chromatin state assessed by ATAC-seq, showed the strongest and the 2nd strongest heritability enrichment with mLOY, respectively (Fig. 3c; Supplementary Table 8 and Supplementary Note 3). We found decreasing heritability enrichment in genes expressed at subsequent differentiation stages; the further differentiated the cell type was from HSC and MPP, the less enrichment was observed. These results suggest that not only HSC but also MPP may be important differentiation stages at which genetic factors influence the development of mLOY.

**Enriched heritability in transcription factor binding sites.** Previous studies have revealed that transcription factors (TFs) play a critical role in hematopoietic cell differentiation and function[27]. Thus, we hypothesized that overlap of genetic associations with TF-binding sites in various hematopoietic cells may provide information about lineage- or context-specific influence on mLOY. We used available chromatin immunoprecipitation sequencing (ChIP-seq) data to help identify TFs involved in mLOY. We

obtained a total of 37 ChIP-seq data sets derived from CD34-positive cells or HSCs (for details, see the Methods section). Binding sites of three transcription factors, namely, FLI1, GATA2, and RUNX1, showed significant overlap with regions positively enriched in mLOY heritability ($P \leq 4.0 \times 10^{-4}$, Fig. 3d). GATA2 and RUNX1 are TFs critical for HSC and hematopoiesis[28,29], supporting the findings above. We focused on binding sites of FLI1, a TF important for megakaryocyte differentiation[30], which showed the strongest heritability enrichment ($P = 1.5 \times 10^{-6}$, P-value for enrichment, and Supplementary Note 3). Binding sites of FLI1 that associated with mLOY heritability were independent from those of GATA2 and RUNX1 ($P = 4.1 \times 10^{-4}$, P-value for enrichment). Heritability enrichment in FLI1-binding sites was not seen in the 11 other cell types for which FLI1 ChIP-seq data were available, supporting the biological specificity of this result; heritability of mLOY is enriched in binding sites of FLI1 in CD34-positive cells (Supplementary Fig. 8). We searched for possible enrichment of mLOY heritability in TF-binding sites for any cell type. Notably, the FLI1-binding sites in CD34-positive cells showed the strongest heritability enrichment across all 2861 TF-cell type pairs available (see the Methods section and Supplementary Fig. 9). Among megakaryocyte/erythrocyte progenitors (MEP), common lymphoid progenitors (CLP) and granulocyte/macrophage progenitors (GMP) (which are similarly differentiated and lineage-committed progenitors, Fig. 3c), the highest heritability enrichment in expressed genes was observed for MEP. This is broadly consistent with heritability enrichment in binding sites of FLI1, which is critical for differentiation of MEP[31]. Importantly, ES cells and iPSC, which have potential to differentiate to HSCs, did not show strong heritability enrichment in TF-binding sites (357 ChIP-seq data, $P > 0.05/357$, P-value for enrichment, Supplementary Fig. 10), emphasizing that interactions in the nucleus of HSCs or hematopoietic progenitor cells which appear after differentiation to the hematopoietic lineage are important for mLOY.

**Associations between hematopoietic traits and mLOY.** FLI1 drives MEP[31] to develop into megakaryocytes and thus produce platelets rather than erythrocytes[32,33]. Therefore, we analyzed the association between mLOY and hematologic traits (including platelet count and red blood cell (RBC) count) using data from 57,987 subjects from whom complete blood counts (CBC) were available at the time of DNA collection. mLOY was positively associated with platelet count (i.e., higher platelet count, lower mLRR-Y, and higher mLOY; 1 SD increase in platelet count (66,000/ul) associated with 0.03 SD decrease in mLRR-Y signal, $P = 7.6 \times 10^{-14}$, Wald test in linear regression analysis) and WBC count ($P = 4.2 \times 10^{-15}$, Wald test in linear regression analysis, Supplementary Fig. 11). In contrast, we observed a nonsignificant negative association with RBC count (i.e., higher RBC count, higher mLRR-Y, and lower mLOY, Supplementary Fig. 11). The effect size of platelet count (and RBC) on mLOY was enhanced in individuals estimated to have a high fraction of cells with loss of chromosome Y (Fig. 3e).

These results are compatible with significant heritability enrichment in FLI1-binding sites. Since RUNX1 also regulates maturation of megakaryocytes and represses erythrocyte gene expression during megakaryocytic differentiation[34], these results also support significant heritability enrichment of RUNX1 binding sites. We did not find significant genetic correlations between mLOY and platelet or RBC counts (Supplementary Table 9 and Supplementary Note 4). This suggests that the association between mLOY and platelet count may be derived from limited genetic components (likely including FLI1 and RUNX1) rather than from diverse polygenic effects or from shared nongenetic factors.

**Pathway analysis and Mendelian randomization of mLOY.** Our results so far suggest that the genetic variants and polygenic associations of mLOY mostly overlap with functional elements in specific hematopoietic progenitor cells. However, the specific cellular pathways and gene-regulatory networks involved in mLOY remain uncertain. To address this, we conducted pathway analysis using PASCAL software[35] and genetic correlation analyses with other quantitative and qualitative traits (see Methods). PASCAL takes not only GWAS significant loci but whole-genome signals into consideration. We found that gene scores calculated based on the GWAS signal of mLOY showed the strongest and most significant enrichment in cell cycle pathways and a mitotic pathway (Supplementary Table 10). This is compatible with the partial overlap of mLOY susceptibility genes and oncogenes[9] as well as with the hypothesis that mitotic clonal expansion of an HSC lacking chromosome Y underlies lower mLRR-Y values. We also conducted genetic correlation analyses between mLOY and quantitative traits or malignancies with which we previously reported genetic associations[36]. We observed a significant genetic correlation between mLOY and aspartate transaminase (AST) (Supplementary Fig. 12 and Supplementary Table 9). A bidirectional Mendelian randomization (MR) approach suggests a causal effect of mLOY on increase of AST ($P = 0.037$, Supplementary Note 4).

**Associations between mLOY and clinical outcomes.** Finally, we investigated the clinical significance of mLOY. We analyzed survival and mLOY in 54,887 BBJ participants followed at a maximum of 12 years after registry[37] (for details, see Methods). There was no association between mLOY and overall survival (Fig. 4a). A previous study that found an association between mLOY and decreased survival used a different analytic approach, one which is strongly influenced by the small number of subjects who have a high fraction of cells lacking chromosome Y (Supplementary Note 5)[3]. Therefore, we tested whether subjects with a high fraction of cells lacking chromosome Y have adverse outcomes in the studied cohort. We found that subjects with lower mLRR-Y values (indicating a higher fraction of cells lacking chromosome Y) were more likely to have experienced an adverse outcome (Supplementary Fig. 13), which seems compatible with the previous finding[3].

Genome instability, including chromosome rearrangements and loss, is a central event in development of some cancers, so genetic determinants that increase susceptibility to mLOY may influence cancer risk. We therefore analyzed whether the associations between mLRR-Y and mortality were mediated by malignances. We found a significant association between mLOY and death due to lung cancer ($P = 0.0010$, Cox proportional hazards regression, Hazard ratio 1.09 (95% CI: 1.04–1.15), Fig. 4b) after conditioning on covariates, including smoking. Subjects with a high fraction of mLOY consistently showed associations with death of lung cancer (Supplementary Fig. 14). However, we found that the association between mLOY and lung cancer was mostly contributed by individuals with a smoking history; the confounding effect of smoking on both lung cancer and mLOY is difficult to disentangle even conditioning on this covariate. When we extracted and analyzed subjects with detailed smoking information including smoking duration and quantity (at the expense of decreasing sample size), the effect size of mLOY on lung cancer was decreased, and the association was no longer significant (Supplementary Note 6). We did not find significant associations between mLOY and other cancer types or with all malignancies (Fig. 4b).

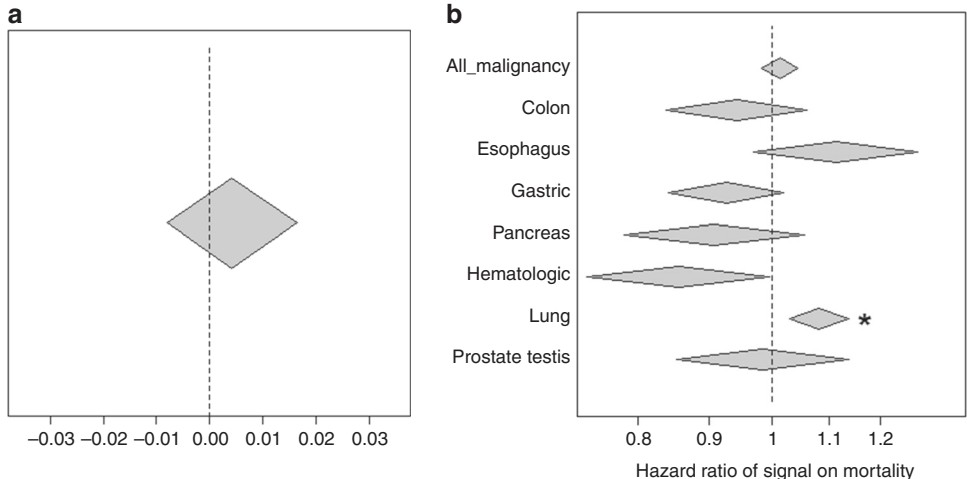

**Fig. 4** Inference of clinical significance of mLOY. **a** Associations between overall mortality and signals of mLOY. **b** Associations between signals of mLOY and mortality due to malignancy and organ-specific malignancy. Signals of mLOY are multiplied by −1 of normalized LRR (the lower LRR indicating large fraction of mLOY, the higher signals are). Thus, beta exceeding 0 and hazard ratio exceeding 1 indicate associations between poor outcome (increased mortality) and lower intensity (more mosaic). Width of diamonds indicates 95% confidence interval. All results are obtained in Cox proportional hazard models. *indicates significant associations beyond Bonferroni's correction

## Discussion

This is a large study to analyze mLOY in a non-European population. In accordance with the large study of mLOY in Europeans (Wright et al.[9]), we used the mean of LRR as a proxy of chromosome Y copy number. Using the mean, rather than the median, of LRR allows that some strongly deviated markers could unduly influence estimation of chromosome Y dosage. However, we found that the median and mean are strongly correlated (Spearman's rho: 0.993) and lead to quite similar genetic association results (median LRR identifies 44 of the 46 loci identified using mean, Supplementary Note 7). The minimal difference in the two indices supports the rationale to use the method applied in this study.

Our analyses provide insights into the mechanisms underlying mLOY, suggesting that MPP and HSC are likely the primary cell types undergoing of clonal expansion after loss of chromosome Y. Genes expressed in hematopoietic lineages, especially HSC and MPP, seem to be critical for mLOY development. As our results suggest variants overlapping myeloid precursor functional regions are involved in the development of mLOY, it would be interesting to analyze subgroups of WBC in elderly populations to identify if myeloid populations contain more cells lacking chromosome Y than co-circulating lymphoid cells.

It is of note that while mLOY was detected by assessing nucleated leukocytes, platelet and RBC counts were associated with mLOY. The involvement was also supported by heritability enrichment of FLI1 and GATA2 bindings. This indicates that the molecular and cellular mechanisms underlying mLOY act before cell fate is restricted to certain terminally differentiated nucleated WBC types, in line with the importance of MPP and HSC shown by enrichment analyses of GWAS significant signals and polygenic signals. The association between WBC counts and mLOY in a nondose dependent manner may reflect higher chance to detect mLOY signals from probe intensity in subjects having more WBC.

Genetic correlation between AST and mLOY and the result of MR suggested a causal relationship between mLOY and increased AST (more mLOY leading to higher AST). Since recent studies have revealed that mosaic events frequently occur systemically[38], this may suggest that mLOY also occurs in the liver and affects susceptibility to liver damage. It is of note that we did not observe an association between mLOY and ALT. This suggests that increase in AST level or ratio of AST and ALT in elderly populations could serve as a marker of mLOY. Especially given the cell-type specificity of the genetic factors we identified, determining the genetic landscape of mLOY in other tissues including the liver would be interesting.

Our study also analyzed association between mLOY and survival at an unprecedented scale. While we could replicate a trend of poor outcome in subjects with a high burden of mLOY, the clinical significance of mLOY was not very apparent in our study. Since this is a large study to analyze effects of mLOY on clinical outcomes in an Asian population, it is possible that the clinical significance of mLOY is not the same across all populations. However, it is also possible that mLOY is associated with onset of specific types of cancer and/or context-dependent survival, such as poor outcome in specific diseases, which could not be fully captured by our population cohort.

While different LD structure between Japanese and UK population made it difficult to quantify genetic similarities of mLOY between the two populations, we showed that significant variants found in the BBJ very often shared direction of effect on mLOY in the UKB. We tried to replicate 156 significant variants found in UKB, as a result, 92 out of the 100 variants which were polymorphic and passed QC showed shared direction in effect[39], suggesting the strong overlap of susceptibility loci for mLOY between the two populations. Considering common genetic background of mLOY, susceptibility loci which are found significant only in one population suggest that the causal variants in these loci are polymorphic only in that population. In addition, taking advantage of genetic overlap seemed to help us to pinpoint shared causal variants of susceptibility loci.

Further in-depth analysis including trans-ethnic meta-analysis could reveal additional genetic components and context-specific clinical significance of mLOY. Because we showed mLOY shares genetic architecture across populations, trans-ethnic meta-analysis would be attractive to increase the number of associated loci and to obtain further biological insights from finer resolution functional inference. The association between mLOY and clinical outcomes should be replicated to resolve the clinical impacts of mLOY and develop management protocols for elderly populations. Disease-specific associations of mLOY should be analyzed in disease-specific data sets rather than population cohorts.

## Methods

**Subjects and genotype data**. We used 183,899 subjects in the BBJ project for selection of samples to analyze for this study. Written informed consent was obtained from all the participants. The study was approved by the ethical committees in the Institute of Medical Science, the University of Tokyo and RIKEN Center for Integrative Medical Science. Most of the subjects had already been genotyped using genome-wide genotyping arrays[40]. In this study, we used genotype data from three different sets using four different arrays, namely, (1) HumanOmniExpressExome v1.0, (2) HumanOmniExpressExome v1.2, and (3) a combination of Illumina HumanOmniExpress v1.0 and Human Exome v1.0 or v1.1 BeadChips. A breakdown of the subjects and arrays is given in Supplementary Table 1.

**Estimation of mLOY**. Using log-R ratio (LRR) generated by Illumina GenomeStudio software, mean LRR in Y chromosome (mLRR-Y) was calculated and used as a proxy of mLOY as follows. We extracted only male subjects and reclustered the subjects based on their probe signal intensities in variants on chromosome Y. This step was performed separately for each type of array. We did not use intensity data of Human Exome BeadChip to avoid batch effects within the same subjects. We obtained LRR for all variants on chromosome Y excluding the pseudoautosomal regions, and subjects and variants with missing rate higher than 5% (restricted to the Y chromosome) were excluded from mLRR-Y calculation. As a result, 1268, 1305, and 1162 variants remained for calculation of mLRR-Y in the three batches described above, respectively. We set LRR in variants with missing genotypes as missing. We then standardized LRR in each variant, and calculated mean of LRR in each subject across variants on chromosome Y. The mLRR-Y was standardized per batch and used as a signal reflecting mLOY in the subjects. We took this approach to avoid strong influence of limited number of variants with extreme values of LRR. To confirm that our data did not suffer from noise due to variants with extreme LRR, we also took median of LRR signals in each individuals among the variants and conducted GWAS with the use of median LRR instead of mLRR-Y.

**Quality control of subjects and SNPs**. We excluded subjects found to be genetically identical to other subjects (321 samples), showing discrepancy between reported sex and inferred sex based on genotypes of variants on X chromosome (1245 samples), found to be outliers from the EAS (Japanese and Chinese) cluster (121 samples) in the analysis where we merged GWAS data with 1000 Genomes Project genotype data, pruned variants by excluding variants in moderate LD with other variants (with cutoff level of $r^2$ of 0.5) and conducted principal component (PC) analysis to project subjects in PC1 and PC2 space, included in the reference panel mentioned below (939 samples) or showing a missing rate higher than 2% (0 sample at this stage). Finally, we obtained data of 95,380 males whose probe intensity data in the Y chromosome was available for subsequent analyses. SNPs with missing rates more than 1%, Hardy–Weinberg equilibrium $P$-values $<1.0 \times 10^{-6}$ or heterozygote calls <5 in each of the three arrays were excluded.

**Whole-genome imputation**. All genome-scanned samples were merged before phasing and whole-genome imputation was conducted using a reference panel which was produced by combining genotype data from 2504 subjects from the 1000 Genomes project[41] and whole-genome sequence data (x30) of 1037 Japanese subjects[16] (see for detail, Supplementary Note 1). Eagle[42] and Minimach3[43] softwares were used for phasing and imputation, respectively.

**Genome-wide association study**. After whole-genome imputation, we tested 9,591,901 variants with squared Pearson correlation ($r^2$) >0.3 (to ensure imputation accuracy) and minor allele frequency >0.005 for association with mLOY. We applied the Bayesian mixed-model using bolt-lmm[44] (URL). Age, arrays, smoking, and disease status (prevalence >0.5% in the subjects) were included as covariates (Supplementary Table 11 and Supplementary Note 8). Since we used mixed model taking into account genetic correlation matrix among subjects, we did not include principal components in the covariates. The mean of smoking status was used for subjects with missing information. $P$-values of $5.0 \times 10^{-8}$ was set as genome-wide significant level.

**Variance and heritability estimation of mLOY**. We calculated variance explained by variants by the following formula:

$$\text{Var\_explained} = 2 \times \text{E}^2 \times \text{af} \times (1 - \text{af}) \times \text{Var\_tot}^{-1}$$

where Var_explained indicates variance explained by a SNP, E indicates beta in LMM, af indicates allele frequency of tested allele, and Var_tot indicates variance of mLOY (we set to 1 by data standardization).

We used bolt-lmm software to evaluate heritability of mLOY. Briefly, bolt calculates heritability of model SNPs (in this case, we provided genotyping array data) by restricted maximum likelihood estimation (REML) to compute variance of genetic components.

**Independent signals in a single locus**. Two associated loci on the same chromosome were regarded as different if a genome-wide significant marker at one locus was at least 1 Mbp apart from those in the other locus. This definition was

also applied to regard significant loci in the BBJ as the same as or different from those described by Wright et al.[9]. We conducted conditional analyses to confirm that two loci were independently associated with mLOY when the boundary of the two regions were <1 Mbp apart. If significant associations were observed in conditional analyses and adjacent loci were apart by >200 kbp but <1 Mbp, we regarded the two loci as different.

**Conditional analysis**. We applied conditional analyses to identify independent signals in a single locus or adjacent loci. We incorporated dosages of a variant whose effect we would like to condition on as a covariate in linear mixed model. We applied the same threshold ($P < 5.0 \times 10^{-8}$) to the conditional analyses to define significant associations.

**Replication of the associations with the UKBB data**. We obtained UKBB association results of the 46 top variants in the BBJ. In the UKBB data, 205,011 male subjects were analyzed, and presence of mLOY was determined by hidden Markov model together with phased B allele frequencies calculated by signal intensity data of markers in pseudo autosomal region (PAR)1 of genotyping arrays[39]. We compared risk allele between the two populations. We applied the same MAF threshold of 0.005 as the BBJ to the UKBB. The four secondary signals in the BBJ were not included, since top variants were different between the two populations and statistics in conditional analyses with the use of the same conditioned variants were not available.

**Functional annotation using Haploreg**. We analyzed overlap between lead variants in significant loci associated with mLOY and enhancer marks in cells by Haploreg (URL) to infer cell types important for mLOY. Briefly, enrichment of lead SNPs for mLOY in enhancer histone marks in cell types was evaluated by comparing estimated overlap based on pruned variants in the 1000 Genomes Project with minor allele frequencies >0.05 in any population. The enrichment significance was computed by binomial test. Since Haploreg requires rs id of SNPs which is not available for Chr2:136879065, we used rs6751768, which is in strong LD with Chr2:136879065 ($r^2 = 0.80$), for this locus. A significant level of associations was set based on Bonferroni's correction.

**Pathway analysis**. We conducted pathway analysis with the use of the PASCAL software[35] (URL) which takes LD structure of nearby genes into consideration. To avoid overestimation of GWAS significant variants, we calculated gene score for pathway analysis by PASCAL by using the sum of statistics of variants in single genes rather than the maximum of the statistics. Bonferroni's correction accounting for all of the pathways (REACTOME, KEGG, and BIOCARTA) was used to set a threshold of statistically significant enrichment ($P < 0.05/1077$).

**LD score regression to estimate confounding bias**. To evaluate the polygenic effect on mLOY and assess confounding bias leading to inflation of median chi-square statistics, we used LD score regression analysis with the use of ldsc software[18]. We regarded intercept in LD score regression <1.05 or ldsc ratio (an index estimating confounding bias of GWAS statistics) <0.3 as no confounding bias[18].

**Partitioning heritability for cell groups and cell types**. We evaluated enrichment of heritability of histone marks in cell groups or detailed cell types in each cell group by conditioning on the full baseline model described in Finucane et al.[23] by ldsc. Briefly, the full baseline model contains coding, intronic, UTR, promoter, enhancer regions, histone marks, open-chromatin regions, and their extended regions, none of which are specific to any cell types[23]. Although ldsc uses chi-square statistics which are calculated from different distribution from that of bolt-lmm[44], we confirmed similar results of ldsc between bolt-lmm chi-square statistics and normal linear regression chi-square statistics (with the same covariates as bolt-lmm). Thus we showed results of analyses with the use of chi-square statistics in bolt-lmm. We also applied to the analyses another full baseline model of ldsc recently reported including information of LD-related annotations, synonymous, and nonsynonymous annotations, ancient sequence age, and conserved function across species[24].

**Genetic correlation**. We also used ldsc software to assess genetic correlation[45] of mLOY and quantitative trait or malignancy susceptibility (Supplementary Note 4). We used 44 blood tests out of the 58 quantitative traits we previously reported[36] and whose summary statistics are available (JENGER, see URL). We used GWAS of malignancy whose summary statistics are available in the BBJ[36] (JENGER, see URL). While samples were overlapped between traits, a genetic correlation estimated by ldsc is shown not to be affected by sample overlap since sample overlap does not change LD score[45].

**Mendelian randomization analysis**. We conducted Mendelian randomization analysis to assess causal relationship between mLOY and AST which showed a significant genetic correlation with mLOY (Supplementary Note 4). We used a Generalized Summary-data-based Mendelian Randomization approach

implemented in GCTA software[46] (bidirectional Mendelian randomization with option –gsmr-direction 2).

**Partitioning heritability with the use of ChIP seq data.** We used the ChIP-seq data in the previous report[47]. Raw human ChIP-seq data files in SRA format were obtained from the GEO database and were converted to FASTQ format using the fastq-dump function of SRA Toolkit. Each sequence read was aligned to the human hg19 genome using Bowtie2 version 2.2.5 with default parameters. Peaks were called using Model-based Analysis of ChIP-Seq (MACS) version 2.1 with default settings ($q < 0.01$). Through this analysis, we obtained a total of 2856 ChIP seq data. LD scores of transcription factor binding were constructed by extending 500 bp from the peaks in ChIP seq. We conducted partitioning heritability of GWAS data of mLOY by ChIP seq using ldsc. We first extracted 37 ChIP seq data, in which HSC was analyzed for transcription factor binding based on the results of enrichment analysis of significant variants in cell-specific enhancer marks and of cell-type or group analyses of partitioning heritability. We partitioned heritability of mLOY with the use of the LD scores of transcription factor bindings and the full baseline model by Finucane et al.[23]. In addition, we also analyzed the data with the use of the latest full baseline model described above.

**LDSC-SEG.** To evaluate the important cell types among the CD34+ cells, we used LDSC-SEG[48], a method recently developed to estimate heritability enrichment of gene expression (or open chromatin) in various tissues and cell types by taking advantage of tissue- and cell-specific eQTL data, which are available in the widest range of tissues and cell types. We used data from Corces et al.[26] who generated ATAC-seq data of hematopoietic cell lineages, including CD34+ hematopoietic stem progenitor cells. We computed Japanese LD scores of cell-specific gene expression based on the ATAC-seq data by referring to European annotations for the analyses (we obtained very similar results regardless of the origin of LD scores). In this analysis, we used the full baseline model by Finucane et al.[23] as baseline of LD scores. In addition, we also analyzed the data with the use of the latest full baseline model described above.

**Survival data in the BBJ.** The living status (dead or alive) of a total of 141,612 BBJ subjects with one of 32 diseases was prospectively followed for more than 10 years after DNA collection[37]. If a death was reported, a detailed search was conducted to identify causes of death coded by ICD10 by accessing the national vital registration system at the Japanese Ministry of Health, Labor, and Welfare. Further details are written elsewhere[37]. We extracted cancers with >1000 deaths overall, and analyzed lung, colon, esophagus, gastric, pancreas, hematologic and prostate/testis cancer as specific malignancies. To analyze associations between mLOY and malignancy mortality, we excluded subjects having had malignancy at registry, without information of follow-up period or with follow-up <1 year to exclude potential undiagnosed cancers and other fatal diseases. As a result, 54,887 subjects with follow-up period of mean 8.0 years and standard deviation of 2.4 years remained for the analyses. During the follow-up periods, 12,410 deaths were observed. We used the survival package of R statistical software. We set subjects who did not die during follow-up as controls and evaluated associations between mLOY and mortality of overall or specific diseases. In addition to the use of mLRR-Y signals as a quantitative trait, we used subjects with the lowest 3, 5, 10, and 20% of mLRR-Y signals as categorical variables and compared with subjects whose mLRR-Y signals not reaching the lowest 20%. The 20% cutoff was determined by the latest UKBB mLOY study which reports 20% of male subjects in the UKBB were positive for mLOY[39]. We used Cox proportional hazards regression for survival analysis to estimate associations between mLOY and the cause of death adjusting for age at DNA collection, data batches, disease status, and smoking.

**Associations between mLOY and CBC data.** We analyzed a total of 57,987 subjects for associations between mLOY and CBC data for inference of biological insights. We z-transformed CBC data to assess associations. Age, smoking, disease status, and data set were used as covariates in the association study. To test the hypothesis that subjects carrying high mLOY signals were strongly associated with CBC data, we took subjects carrying top 1, 3, 5, and 10% of mLOY signal (i.e., those with the lowest mean LRR of variants in chromosome Y). We divided all subjects in this study into two groups (binary traits), with or without top 1, 3, 5, and 10% of mLOY signals and associated the binary traits with the CBC data mentioned above together with covariates of age, smoking, disease status, data set with the use of logistic regression analysis.

**URL.** For 1000 Genomes Project, see http://www.1000genomes.org/; for GWAS catalog, see https://www.ebi.ac.uk/gwas/; for LDSC and SEG-LDSC, see https://github.com/bulik/ldsc/; for PASCAL, see https://www2.unil.ch/cbg/index.php?title=Pascal; for Minimac, see https://genome.sph.umich.edu/wiki/Minimac; for Haploreg, see https://pubs.broadinstitute.org/mammals/haploreg/haploreg.php; for JENGER, see http://jenger.riken.jp/en/; for R, see https://www.r-project.org/; for Plink1.9, see https://www.cog-genomics.org/plink2; for bolt-lmm, see https://data.broadinstitute.org/alkesgroup/BOLT-LMM/; for LCR, see ftp://ftp.1000genomes.ebi.ac.uk/vol1/ftp/release/20130502/supporting/low_complexity_regions/hs37d5-LCRs.20140224.bed.gz.

**Reporting summary.** Further information on research design is available in the Nature Research Reporting Summary linked to this article.

## Data availability
GWAS summary statistics of mLOY in the BBJ is available at RIKEN website (http://jenger.riken.jp/en/). While individual-level genetic data are not accessible, all other data contained in the article and its supplementary information are available upon reasonable request.

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

## Acknowledgements

We deeply thank Dr. Nicholas Parrish for critically reviewing and editing the manuscript. We thank all of the staff in the BBJ project for their efforts to keep and handle samples. This research was conducted using the UK Biobank Resource under Application #19808. P.-R.L. was supported by NIH grant DP2 ES030554, a Burroughs Wellcome Fund Career Award at the Scientific Interfaces, the Next Generation Fund at the Broad Institute of MIT and Harvard, and a Glenn Foundation for Medical Research and AFAR Grants for Junior Faculty award. G.G. was supported by US Department of Defense Breast Cancer Research Breakthrough Award W81XWH-16-1-0316 and the Stanley Center for Psychiatric Research.

## Author contributions

C.T. and K.Y. designed the study. C.T. analyzed the data. C.T., P.R.L., G.G., J.P. and Y.K. wrote the paper. P.R.L., G.G. and J.P. contributed to the UKB association results. K.I., E.K., H.S. and T.O. made data of TF-binding sites. Y.M., M.H., K.M., Y.M., M.K. and Y.K. contributed to creation of the BBJ genetic data, clinical information, and follow-up data. K.I. and M.A. contributed to make an imputation reference panel. All authors critically reviewed and approved the final version of the paper.

## Competing interests

The authors declare no competing interests.
