## [Peer Review File · Nature Communications]

Reviewers' Comments:

Reviewer #1:

Remarks to the Author:

Terao et al present an analysis of mosaic loss of Y using a data set of 95 thousand Japanese males. They present a GWAS for genetic predictors of acquired Y loss, using integrated molecular epidemiology techniques. While aspects of their study were modern, and the importance of documenting loss of Y (or other chromosomes) is novel in this East Asian population, the presentation of details was poor, diminishing the impact of the paper.

Major Points

Pages/lines were not numbered making it difficult to reference the comments below.

1) This manuscript read as if it were a first draft. The authors need to edit both for grammar and for clarity. Please focus on using precise language. The manuscript is missing a discussion section (if format doesn't permit at least a discussion paragraph needs to be added)?

Overall there is a lack of clarity in exposition. I cannot judge the science in some sections because information is missing.

Abbreviations must be explained in each section or at least the paper (they cannot be introduced only in the Abstract)

That one variant is not polymorphic in the Japanese population.... "Given the observed strong overlap... suggests that a causal variant in Europeans might not be polymorphic..." → this comment is inane given what was established. What is the additional point here? Seemed highly repetitive.

"The new loci found in the current study... showed overlap of HSC." → ??? "overlap of HSC" is not informative whatsoever and it is not grammatically correct. The last sentence of this paragraph was extremely difficult to parse — what point are the authors trying to make? There are not enough details and it is lacking a good explanation.

"We found expressing genes supported by open chromatin....". This sentence is poorly written and confusing for the reader. The authors may have an important point but it's lost.

2) The authors used median and average LRR and get slightly different results (two loci differ), they need to investigate this discrepancy and justify why they used average LRR. → This is highly descriptive and what does it mean? If one says "we looked at both" and then picked one, it comes across as fishing. (In my opinion they should use median LRR, instead of average LRR for the subsequent analyses.)

3) The authors need to provide more detail regarding their methods. This is a mandatory point for serious consideration of the manuscript. It is impossible to give a full account of examples but for one:

.... mLOY with the use of the 53 "basic annotations".... What basic annotations? A few were referenced. But this is assuming the audience for this paper can read the mind of the authors.

There are numerous instances such as this.

4) The authors need to frame motivation for analyses better. They don't frame their overall analysis plan. They just jump from one integrated genetic epi analysis to another.

5) The authors did a nice job with the analysis of survival.

Some minor comments

Right at beginning of Results... Age and mLOY. Figure 1 does not show biological interpretation. What proportion of individuals exhibit loss? To what degree is loss exhibited in those individuals with loss? The combined result is highly significant but $P < 1e-100$ is uninformative beyond a demonstration of at least non-zero effect and sample size.

Smoking. Again, what is the effect? It's the first paragraph but buried in the Supplement in a Figure for the reader to figure out.

Reviewer #2:

Remarks to the Author:

Terao et al. performed the first also the largest GWAS of mosaic loss of chromosome Y (mLOY) in East Asian population, using genetic information from 95,380 Japanese males, and mean logR-ratio as a proxy for the phenotype mLOY. The authors identified 50 independent genetic markers in 46 loci, of which, 35 were novel. They also performed a compressive function annotation analysis, using Haploreg, PASAL pathway analysis, LD score regression, LDSC-SEG and more algorithms.

Despite a large amount of results, and different layers of data (e.g., functional annotations, survival data, environmental exposures such as smoking, hematological traits, etc), the manuscript is to some extent, not so very well-organized.

1. As an original article (at least from my end, the manuscript was submitted as an original article not a letter), the manuscript lacks discussion, which should have been there to provide interpretations and implications to the current results.
2. A proper summarization or conclusion is also lacking.
3. Although mendelian randomization (MR between mLOY and AST) was mentioned in the methods, it was never cited in the results (supplementary figure 14 & 15) – indeed those relevant parts can be found in supplementary note, authors could at least be able to mention “see supplementary notes” instead of dropping them entirely from the main text.
4. Repeated presentations to the same results, which did not add extra information or values – for example supplementary table 14 and supplementary figure 13 were actually the same thing – I would suggest the authors keep only one of these two items given that these results do not add extra values to each other. I would also suggest the authors drop short tables which has only 1 row such as supplementary table 10 – it is sufficient to mention this piece of result using a short sentence. (please correct through the whole manuscript to avoid such situations – write clearly and succinctly, not necessarily piling up all the results).
5. Some sentences read ambiguous and confusing, for example, “The highest heritability enrichment in expressing gene of megakaryocyte/erythrocyte progenitor (MEP) among MEP, common lymphoid progenitor (CLP) and granulocyte/macrophage progenitor (GMP) (the similar differentiation stage after HSC, Figure 3C) might also be compatible with enrichment in FLI1 binding sites which is critical for MEP)”, “Subjects with high fraction of mLOY consistently showed associations (of what?)”, “This trend was consistent when restricting to the 31 novel loci, with 23 out of 25 variants having the same effect direction in the two studies (where comes from the 25 variants when the sentence began with 31 loci?)”, “Within contrast, we observed a non-significant negative association with RBC count”, etc. I would suggest a universal language editing to the manuscript.

In addition to those major points, there are several minor comments:

1. For the GWAS analysis, age, array, smoking and disease status were included as covariates. Wonder if principal components (say, first 2 PCs) were also included to control for population stratification?
2. 53 basic annotations: Hilary Finucane's annotation vs. Steven Gazal's annotation. The author may try Gazal's [PMID: 28892061] baseline model which takes into consideration the LD pattern.
3. Genetic correlation between mLOY and hematologic traits, since all these traits were based on the same bunch of individuals (BBJ), have the authors accounted for sample overlapping?
4. Results from BBJ were replicated by using UKBB data – wonder how similar is the phenotype (mLOY) between UKBB vs. BBJ, given that the two cohorts used different algorithms to calculate the proxy? Maybe quantify the genetic correlation between the UKBB mLOY (estimated using hidden Markov model) and BBJ (log-RR) mLOY.
5. A small typo: "We found that mLOY is a highly polygenic trait and departure of mean chi-squared statistics could be largely explained by polygenic effects (lambda genomic control 1.086 > intercept 1.019 in LD score regression, Figure 1B)." – should be 1.066.

We thank the reviewers for taking the time to review our manuscript. We have considered each of the comments carefully, and have strived to improve the manuscript based on their comments.

Aside from response to the comments from the reviewers, we modified our manuscript to strictly follow the journal format. We modified the title of the manuscript.

Reviewers' comments:

Reviewer #1 (Remarks to the Author):

Terao et al present an analysis of mosaic loss of Y using a data set of 95 thousand Japanese males. They present a GWAS for genetic predictors of acquired Y loss, using integrated molecular epidemiology techniques. While aspects of their study were modern, and the importance of documenting loss of Y (or other chromosomes) is novel in this East Asian population, the presentation of details was poor, diminishing the impact of the paper.

> Thank you for your comment. We are very sorry that the manuscript was not well-organized. We intensively modified the manuscript to improve the presentation.

Major Points

Pages/lines were not numbered making it difficult to reference the comments below.

> We are sorry for this. We put line and page numbers in the revision.

1) This manuscript read as if it were a first draft. The authors need to edit both for grammar and for clarity. Please focus on using precise language. The manuscript is missing a discussion section (if format doesn't permit at least a discussion paragraph needs to be added)? Overall there is a lack of clarity in exposition. I cannot judge the science in some sections because information is missing.

>Thank you. We have made efforts to improve clarity. The lack of discussion section was a consequence of transferring this manuscript directly to Nature Communications from

another journal. In revision, we substantially restructured the manuscript to make our arguments more clear and added a discussion section. We also asked native speakers to carefully check the manuscript.

Abbreviations must be explained in each section or at least the paper (they cannot be introduced only in the Abstract)

>Thank you for pointing this out. We carefully reviewed the revision and made modifications. We ensured that abbreviations were explained when they appeared for the first time in the main text.

That one variant is not polymorphic in the Japanese population... "Given the observed strong overlap... suggests that a causal variant in Europeans might not be polymorphic..." -> this comment is inane given what was established. What is the additional point here? Seemed highly repetitive.

>Thank you for pointing this out. We agree that the sentences are repetitive. We deleted the latter part and modified the former part as follows.

(page 6 line 169-)

The top variants in the other three regions identified in Wright et al. 2017 but not here were also very rare or not polymorphic in the Japanese population (Supplementary Table 3). Therefore relative invariance at these 4 loci in the Japanese population precludes identification of a statistically significant signal in this population.

We also carefully reviewed the revision to avoid the similar situation.

"The new loci found in the current study... showed overlap of HSC." -> ??? "overlap of HSC" is not informative whatsoever and it is not grammatically correct. The last sentence of this paragraph was extremely difficult to parse – what point are the authors trying to make? There are not enough details and it is lacking a good explanation.

>Thank you for the comment. Please excuse us for the poor presentation of concepts in this paragraph. We carefully reviewed the revision to improve grammar and scientific clarity. The parts pointed out by the reviewer were modified as follows to make arguments clear.

(page 7 line 194~)

The 31 new loci found in the current study accounted for this overlap. Two completely independent sets of variants (31 and 19 loci in Japan and UK, respectively) showed overlap with enhancer marks in HSC (Supplementary Figure 7). Enhancer marks in embryonic stem cells (ES cells) or induced pluripotent stem cells (iPS cells), both of which can differentiate into HSC²¹, did not strongly overlap mLOY GWAS significant signals. Thus the variants associated with mLOY overlap with cell type-specific epigenetic features emerging after differentiation into hematopoietic cells, rather than epigenetic features characterizing pluripotency itself.

"We found expressing genes supported by open chromatin...". This sentence is poorly written and confusing for the reader. The authors may have an important point but it's lost.

> Thank you for the comment. We modified this part accordingly as shown below. In addition, we carefully reviewed the revision to avoid similar situation.

(Page 8 line 237~)

Genes expressed in MPP and HSC, evidenced by open chromatin state assessed by ATAC-seq, showed the strongest and the 2nd strongest heritability enrichment with mLOY, respectively (Figure 3C, Supplementary Table 8 and Supplementary Note 3).

2) The authors used median and average LRR and get slightly different results (two loci differ), they need to investigate this discrepancy and justify why they used average LRR. → This is highly descriptive and what does it mean? If one says "we looked at both" and then picked one, it comes across as fishing. (In my opinion they should use median LRR, instead of average LRR for the subsequent analyses.)

> Thank you for the comment. In the revised manuscript, we follow the precedence of Wright et al in using mean LRR. We show that using median LRR gives a qualitatively similar result, addressing the possibility that some variants with extreme LRR deviated the mean LRR. We confirmed that mean LRR was highly correlated with median LRR and genetic association results were quite similar between the analyses using mean and median LRR. We modified accordingly as follows.

(page 11 line 352~)

In accordance with the largest study of mLOY in Europeans (Wright et al, 2017), we used the mean of LRR as a proxy of chromosome Y copy number. Using the mean, rather than the median, of LRR allows that some strongly deviated markers could unduly influence estimation of chromosome Y dosage. However, we found that the median and mean are strongly correlated (Spearman's rho:0.993) and lead to quite similar genetic association results (median LRR identifies 44 of the 46 loci identified using mean, Supplementary Note 7). The minimal difference in the two indices supports the rationale to use the method applied in this study.

(page 14 line 455~)

We took this approach to avoid strong influence of limited number of variants with extreme values of LRR. To confirm that our data did not suffer from noise due to variants with extreme LRR, we also took median of LRR signals in each individuals among the variants and conducted GWAS with the use of median LRR instead of mLRR-Y.

We also put Supplementary Note 7 to discuss this point.

3) The authors need to provide more detail regarding their methods. This is a mandatory point for serious consideration of the manuscript. It is impossible to give a full account of examples but for one:

... mLOY with the use of the 53 "basic annotations".... What basic annotations? A few were referenced. But this is assuming the audience for this paper can read the mind of the authors.

There are numerous instances such as this.

> Thank you very much for the comment. We fully agree that detailed explanation is important to make ourselves clear in the text.

The basic annotations were previously described as the full baseline model in the original ldsc paper by Finucane et al. The full baseline model contains coding, intronic, UTR, promoter, enhancer regions, histone marks, open chromatin regions and their extended regions, none of which are specific to any cell types. Again, we carefully reviewed the manuscript and intensively modified it to avoid insufficient explanations especially of our methods (we also made Supplementary Notes to address this issue). The sentences

pointed out by the reviewer were modified as follows.

(Page 16 line 567~)

Partitioning heritability for cell groups and cell types.

We evaluated enrichment of heritability of histone marks in cell groups or detailed cell types in each cell group by conditioning on the full baseline model described in Finucane et al, 2015²² by ldsc. Briefly, the full baseline model contains coding, intronic, UTR, promoter, enhancer regions, histone marks, open chromatin regions and their extended regions, none of which are specific to any cell types²².

4) The authors need to frame motivation for analyses better. They don't frame their overall analysis plan. They just jump from one integrated genetic epi analysis to another.

>Thank you very much for pointing out this. We added sentences or phrases explaining the motivation for each analysis to begin each paragraph in the Results section.

5) The authors did a nice job with the analysis of survival.

>Thank you very much for the positive comment.

Some minor comments

Right at beginning of Results... Age and mLOY. Figure 1 does not show biological interpretation. What proportion of individuals exhibit loss? To what degree is loss exhibited in those individuals with loss? The combined result is highly significant but $P < 1e-100$ is uninformative beyond a demonstration of at least non-zero effect and sample size.

>Thank you for the comment. We agree that description of quantitative effect of age is meaningful. We put a description about an effect size of age on mLRR-Y signal in the Result section as written below. Since we analyzed mLOY in a quantitative manner, it is not possible to infer the proportion of subjects carrying the loss. Instead, we showed a quantitative effect of age on mLRR-Y signals.

(Page 5 line 123~)

We observed a strong association between age at DNA collection and mLOY (1

year increase in age associated with 2.2% standard deviation (SD) decrease in mLRR-Y signal, $p < 1.0 \times 10^{-100}$ in multiple linear regression, Figure 1A and Supplementary Figure 1) explaining 9.6% of the variance in mLOY.

Smoking. Again, what is the effect? It's the first paragraph but buried in the Supplement in a Figure for the reader to figure out.

>Thank you for the comments. We modified the manuscript to mention an effect size of smoking on mLRR-Y signals in the result section.

(Page 5 line 126~)

We also observed a significant association between smoking and mLOY (Supplementary Figure 2); smokers often have lower mean intensity of chromosome Y probes, indicating a higher fraction of cells with loss of chromosome Y (smokers associated with 4.6% SD decrease in mLRR-Y signal, $p = 7.5 \times 10^{-10}$). These associations are in agreement with previous studies⁵.

Reviewer #2 (Remarks to the Author):

Terao et al. performed the first also the largest GWAS of mosaic loss of chromosome Y (mLOY) in East Asian population, using genetic information from 95,380 Japanese males, and mean logR-ratio as a proxy for the phenotype mLOY. The authors identified 50 independent genetic markers in 46 loci, of which, 35 were novel. They also performed a compressive function annotation analysis, using Haploreg, PASAL pathway analysis, LD score regression, LDSC-SEG and more algorithms.

Despite a large amount of results, and different layers of data (e.g., functional annotations, survival data, environmental exposures such as smoking, hematological traits, etc), the manuscript is to some extent, not so very well-organized.

1. As an original article (at least from my end, the manuscript was submitted as an original article not a letter), the manuscript lacks discussion, which should have been there to provide interpretations and implications to the current results.

> We are very sorry for this. It reflects the direct transfer of our manuscript from Nature

Genetics in a format lacking discussion. In the revised manuscript, we added a discussion section. We also added detailed explanation of our methods and intensively modified the manuscript.

2. A proper summarization or conclusion is also lacking.

> This relates to the point raised above. We have addressed this by adding conclusions in the revised manuscript.

3. Although mendelian randomization (MR between mLOY and AST) was mentioned in the methods, it was never cited in the results (supplementary figure 14 & 15) - indeed those relevant parts can be found in supplementary note, authors could at least be able to mention "see supplementary notes" instead of dropping them entirely from the main text.

> Thank you very much for pointing this out. We totally agree that we should mention this analysis. We modified the main text accordingly to mention this point as follows.

(Page 10 line 313-)

We also conducted genetic correlation analyses between mLOY and quantitative traits or malignancies with which we previously reported genetic associations³⁵. We observed a significant genetic correlation between mLOY and aspartate transaminase (AST) (Supplementary Figure 14 and Supplementary Table 9). A bidirectional Mendelian randomization (MR) approach suggests a causal effect of mLOY on increase of AST ($p=0.037$, Supplementary Note 4).

4. Repeated presentations to the same results, which did not add extra information or values - for example supplementary table 14 and supplementary figure 13 were actually the same thing - I would suggest the authors keep only one of these two items given that these results do not add extra values to each other. I would also suggest the authors drop short tables which has only 1 row such as supplementary table 10 - it is sufficient to mention this piece of result using a short sentence. (please correct through the whole manuscript to avoid such situations - write clearly and succinctly, not necessarily piling up all the results).

> Thank you very much for the valuable comments. We modified these parts accordingly and carefully restructured the manuscript. As a result, we deleted a total of five supplementary tables which were redundant with Figures.

5. Some sentences read ambiguous and confusing, for example, "The highest heritability enrichment in expressing gene of megakaryocyte/erythrocyte progenitor (MEP) among MEP, common lymphoid progenitor (CLP) and granulocyte/macrophage progenitor (GMP) (the similar differentiation stage after HSC, Figure 3C) might also be compatible with enrichment in FLI1 binding sites which is critical for MEP)", "Subjects with high fraction of mLOY consistently showed associations (of what?)", "This trend was consistent when restricting to the 31 novel loci, with 23 out of 25 variants having the same effect direction in the two studies (where comes from the 25 variants when the sentence began with 31 loci?)", "Within contrast, we observed a non-significant negative association with RBC count", etc. I would suggest a universal language editing to the manuscript.

> Thank you very much for pointing these out. We asked native speakers to carefully check the manuscript again and modified the text to avoid ambiguous expressions throughout the manuscript. The sentences pointed out by the reviewer were modified as follows.

(Page 8 line 240~)

We found decreasing heritability enrichment in genes expressed at subsequent differentiation stages; the further differentiated the cell type was from HSC and MPP, the less enrichment was observed. These results suggest that not only HSC but also MPP may be important differentiation stages at which genetic factors influence the development of mLOY.

(Page 11 line 339~)

Subjects with a high fraction of mLOY consistently showed associations with death of lung cancer (Supplementary Figure 13).

(Page 6 line 180~)

When considering only the 31 novel loci, this trend remained consistent; twenty five variants were polymorphic and had MAF more than 0.5% in the UK population and

23 out of the 25 variants had the same effect direction in the two studies (binomial $P=1.9 \times 10^{-5}$), indicating strong genetic overlap in association with mLOY between the two populations.

(Page 9 line 287~)

In contrast, we observed a non-significant negative association with RBC count (i.e., higher RBC count, higher mLRR-Y and lower mLOY, Supplementary Figure 11).

In addition to those major points, there are several minor comments:

1. For the GWAS analysis, age, array, smoking and disease status were included as covariates. Wonder if principal components (say, first 2 PCs) were also included to control for population stratification?

Thank you very much for this comment. Since we used bolt-lmm in our analyses, we did not use PCs as covariates. Bolt-lmm allows a mixed effect model that takes the genetic relation matrix (GRM) into consideration to compute statistics. We agree that PCs also capture genetic correlations of the data, and our approach using bolt-lmm to take these effects into consideration. Perhaps reflecting the relative genetic homogeneity of our population relative to those in which PCs capture more genetic correlations, when we included PCs in covariates, we obtained quite similar results. We added an explanatory sentence to mention this point.

(Page 15 line 493~)

Since we used mixed model taking into account genetic correlation matrix among subjects, we did not include principal components in the covariates.

2. 53 basic annotations: Hilary Finucane's annotation vs. Steven Gazal's annotation. The author may try Gazal's [PMID: 28892061] baseline model which takes into consideration the LD pattern.

Thank you very much for pointing out this. In addition to the baseline model in the manuscript, we applied the latest baseline model to the enrichment analyses. We again found that superenhancers showed the strongest heritability enrichment. We modified the manuscript to mention this point and make Supplementary Note 3.

(Page 8, line 221~)

The mLOY heritability enrichment in superenhancers, histone marks in the hematopoietic cell group and CD34-positive cells were confirmed in the analyses with ldsc where we used different baseline annotations recently proposed to take into consideration difference in LD pattern of variants, information of synonymous/non-synonymous coding variants, sequence age and conservation of variants across species²³ (see Methods and Supplementary Note 3).

(Page 17, line 577~)

We also applied to the analyses another full baseline model of ldsc recently reported including information of LD-related annotations, synonymous and non-synonymous annotations, ancient sequence age, and conserved function across species²³.

3. Genetic correlation between mLOY and hematologic traits, since all these traits were based on the same bunch of individuals (BBJ), have the authors accounted for sample overlapping?

>Thank you very much for this important comment. We did not consider sample overlap for this analyses since Finucaine et al revealed that sample overlap does not matter for genetic correlation estimation by ldsc since sample overlap do not affect LD score and its beta. We added a description about this point to the Method section.

(Page 17 line 587~)

While samples were overlapped between traits, a genetic correlation estimated by ldsc is shown not to be affected by sample overlap since sample overlap does not change LD score⁴⁴.

4. Results from BBJ were replicated by using UKBB data - wonder how similar is the phenotype (mLOY) between UKBB vs. BBJ, given that the two cohorts used different algorithms to calculate the proxy? Maybe quantify the genetic correlation between the UKBB mLOY (estimated using hidden Markov model) and BBJ (log-RR) mLOY.

>Thank you very much for this comment. While summary statistics of mLOY associations in the UKBB are not available, we tried to replicate 156 significant variants found in UKBB. As a result, 100 variants were polymorphic in Japanese and passed QC. Among the 100

variants, 92 variants shared alleles in effect (Thompson, Biorxiv). We mentioned this point in the Discussion as follows.

Page 12 line 403~

While different LD structure between Japanese and UK population made it difficult to quantify genetic similarities of mLOY between the two populations, we showed that significant variants found in the BBJ very often shared direction of effect on mLOY in the UKB. We tried to replicate 156 significant variants found in UKB, as a result, 92 out of the 100 variants which were polymorphic and passed QC showed shared direction in effect³⁸, suggesting the strong overlap of susceptibility loci for mLOY between the two populations.

5. A small typo: "We found that mLOY is a highly polygenic trait and departure of mean chi-squared statistics could be largely explained by polygenic effects (lambda genomic control 1.086 > intercept 1.019 in LD score regression, Figure 1B)." - should be 1.066.

> Thank you very much for the comment. Sorry for the confusion. As the reviewer indicated, the lambda of the entire GWAS is 1.066. However, LD score regression analysis took variants in Hapmap3 to ensure accuracy of genotypes. This resulted in slight difference in lambdas between original GWAS and LD score regression. We added a Supplementary Note 2 to clarify this point.

Supplementary Note 2. Difference in lambda GC between LDSC regression

and nominal associations.

Since LDSC took variants whose genotypes were highly accurate (in the current analyses, we adopted variants in the hapmap3 project) to ensure accurate genotypes, we observed slight difference in lambda GC between LDSC regression and nominal associations (1.086 and 1.066, respectively).

Reviewers' Comments:

Reviewer #2:

Remarks to the Author:

The authors have adequately addressed my concerns.